# Exploiting the Macrophage Production of IL-12 in Improvement of Vaccine Development against *Toxoplasma gondii* and *Neospora caninum* Infections

**DOI:** 10.3390/vaccines10122082

**Published:** 2022-12-06

**Authors:** Ragab M. Fereig, Mosaab A. Omar, Abdullah F. Alsayeqh

**Affiliations:** 1Department of Animal Medicine, Faculty of Veterinary Medicine, South Valley University, Qena 83523, Egypt; 2Department of Parasitology, Faculty of Veterinary Medicine, South Valley University, Qena 83523, Egypt; 3Department of Veterinary Medicine, College of Agriculture and Veterinary Medicine, Qassim University, Buraidah 51452, Saudi Arabia

**Keywords:** *N. caninum*, *T. gondii*, vaccine, neosporosis, macrophages, antigen

## Abstract

Toxoplasmosis and neosporosis are major protozoan diseases of global distribution. *Toxoplasma gondii* is the cause of toxoplasmosis, which affects almost all warm-blooded animals, including humans, while *Neospora caninum* induces neosporosis in many animal species, especially cattle. The current defective situation with control measures is hindering all efforts to overcome the health hazards and economic losses of toxoplasmosis and neosporosis. Adequate understanding of host-parasite interactions and host strategies to combat such infections can be exploited in establishing potent control measures, including vaccine development. Macrophages are the first defense line of innate immunity, which is responsible for the successful elimination of *T.gondii* or *N. caninum*. This action is exerted via the immunoregulatory interleukin-12 (IL-12), which orchestrates the production of interferon gamma (IFN-γ) from various immune cells. Cellular immune response and IFN-γ production is the hallmark for successful vaccine candidates against both *T. gondii* and *N. caninum*. However, the discovery of potential vaccine candidates is a highly laborious, time-consuming and expensive procedure. In this review, we will try to exploit previous knowledge and our research experience to establish an efficient immunological approach for exploring potential vaccine candidates against *T. gondii* and *N. caninum*. Our previous studies on vaccine development against both *T. gondii* and *N. caninum* revealed a strong association between the successful and potential vaccine antigens and their ability to promote the macrophage secretion of IL-12 using a murine model. This phenomenon was emphasized using different recombinant antigens, parasites, and experimental approaches. Upon these data and research trials, IL-12 production from murine macrophages can be used as an initial predictor for judgment of vaccine efficacy before further evaluation in time-consuming and laborious in vivo experiments. However, more studies and research are required to conceptualize this immunological approach.

## 1. Overview

There is a ‘‘proof-of-concept” that vaccination is the cornerstone for the preventive and control strategies against numerous infectious diseases. In the case of endemic or pandemic situations, vaccination will be the optimal option for reducing hazards and resuming a normal lifestyle. Except for malaria, many protozoan infections are neglected, and the available information is extremely scanty not only for ordinary people but also for some members of the scientific community. *Toxoplasma gondii* (*T. gondii*) and *Neospora caninum* (*N. caninum*) are very similar protozoan parasites in many aspects of structure, biology, and pathogenesis. However, recent advances in research technologies have also revealed many differences, particularly in the genetic makeup. Although such parasites induce tremendous economic losses, no effective treatment or potent vaccine is currently available. The vaccine development for these parasites faces many challenges and obstacles. One of these challenges is the high cost and long time required to find potent vaccine candidates, even at experimental levels. Recently, bioinformatic analysis software programs can predict the antigenicity and immunogenicity of antigens or peptides based on sequence analysis. However, this approach remains far from massive application because of realistic or scientific concerns. Thus, the establishment of an immunoscreening method for the prediction of potent vaccine antigens using an efficient and persuasive model is critically needed. In this review, we attempt to provide an in vitro immunoscreening approach for the discovery of potent vaccine antigens using macrophages; the professional phagocytes and antigen-presenting cells. This system is provided based on summarizing our previous research on vaccine development against *T. gondii* and *N. caninum* in murine models. This valuable information might help vaccinologists in developing potent vaccines against different pathogens in a considerably shorter time and with less effort than traditional procedures. In addition, this information may encourage other researchers to develop different immunoscreening methods based on other immune effector cells or molecules.

## 2. Introduction

*Toxoplasma gondii* and *Neospora caninum* are very similar protozoan parasites in many structural and biological characteristics. Toxoplasmosis and neosporosis, respectively, are responsible for major losses in the medical and veterinary sectors. *T. gondii* may infect practically all warm-blooded animals, with sheep, pigs, and humans being the most vulnerable. Infection is spread mostly through the ingestion of oocysts or tissue cysts. The infection is mostly serious in immunocompromised patients, such as those who are young, old aged, or pregnant women or animals [1,2,3]. Recently, different species of marine mammals are found susceptible to *T. gondii* which has the risk of spreading *T. gondii* oocysts via contaminated water and can result in foodborne *T. gondii* infections in humans [4].

Neosporosis is caused by *N. caninum,* the apicomplexan intracellular parasite. Cattle, sheep, and dogs are severely infected by *N. caninum*. Infection can spread orally via oocysts or tissue cysts, or vertically via the placenta from an infected dam to the foetus. Abortion and culling of infected animals are the main causes of the enormous economic losses in the cattle industry [5,6].

Dogs as definitive host for *N. caninum*, and cats as the definitive host for *T. gondii* are critical for the transmission and epidemiology of neosporosis and toxoplasmosis, respectively, among other susceptible animals. Once infected, dogs and cats might remain a permanent source of infection for other animals. Because of the complexity of the lifestyle of both parasites, successful and potent treatment regimens have not been available to date. Moreover, lacking the potent vaccines, good animal husbandry practices, routine testing of the animals, and estimation of current epidemiological statuses are recommended for prevention of the infection [2,5].

In cases of toxoplasmosis or neosporosis, a primary infection in pregnant sheep and goats, or cattle, respectively, can establish a placental and fetal infection that may result in fetal death, abortion, or stillbirth [7,8,9,10]. As mentioned earlier, abortion induced by the above-mentioned infections is the major and direct cause of substantial economic losses in the livestock production sector. In dairy and meat sheep farms in Spain, the economic losses were obvious due to outbreaks of *Toxoplasma* abortions. Although there were extensive investigations, *T. gondii* was the only and definitive cause of abortion in such sheep flocks. The calculated total direct economic losses were €5154.5 (€171.8/abortion) in the dairy flock and €4456 (€63.6/abortion) in the meat flock [11]. In the case of *N. caninum*, although neosporosis has been reported as a common cause of abortion in cattle, it has also been detected as a cause of abortion in sheep [12]. *N. caninum* can be congenitally transmitted in naturally infected sheep and goats and can cause abortion and perinatal mortality [10,13].

Regarding human infection, approximately 25% of all people could have *T. gondii*. Humans can contract the disease by consuming tainted food and water that includes oocysts [14]. Congenital infection can also develop through vertical transmission, when a woman contracts the disease while carrying a child. Toxoplasmosis causes a relatively mild infection in healthy adults, including a high temperature, swollen lymph nodes, and muscle weakness. Congenital infections carry the risk of more serious results, and the growing foetus may exhibit symptoms that range from severe (often brought on by infection in the first trimester) to moderate (more common when infection occurs later during pregnancy). Intense forms of congenital toxoplasmosis can cause cerebral calcification, hydrocephaly, and microcephaly, intracranial calcification, and even loss of life [15]. Severe infections in immunocompromised people often arise from the recurrence of a persistent illness, as is the case with AIDS, organ transplant, or chemotherapy patients. All these severe consequences are consistent with a lack of long-lasting immunity and the parasite’s capacity to reemerge from tissue cysts and transform back into quickly expanding tachyzoites, which causes tissue destruction [16]. Consistently, specific antibodies to *N. caninum* have been detected in women’s sera using IgG [17] or IgM [18]. However, no evidence for a clinical form of neosporosis has been detected. A recent study, however, discovered that two samples (1%) out of 201 examined human umbilical cord blood samples were Nc5 PCR-positive for *N. caninum*; nonetheless, the placenta tested negative for this parasite [19].

Regarding ultrastructure and pathobiology, both *T. gondii* and *N. caninum* are very similar phenotypically in all developing stages, including tachyzoites, bradyzoites, and oocysts [20,21]. Thus, identification via light or electron microscopy is very difficult, and using highly specific molecular and antigen-antibody detection tools is recommended (Figure 1). Not only do they share phenotypic similarities, but both parasites also possess several similarities in aspects of infection and pathogenesis. These agents are intracellular microorganisms that occasionally evade macrophage killing and take advantage of its multiplication and transmission in order to establish a successful infection and spread to target tissues. In the case of *T. gondii*, avirulent *Toxoplasma* parasites are initially phagocytosed by macrophages and subsequently form a parasite vacuole in a phagosomal compartment. This phagosome to vacuole invasion (PTVI) pathway is associated with a more efficient infection of macrophages [22]. For *N. caninum*, including both Nc-Spain7 and Nc-Spain1H, isolates actively invaded, survived, and replicated in the bovine macrophages, with a higher tendency for the Nc-Spain7 high virulence isolate. However, the molecular interaction responsible for *Neospora* survival and replication in macrophages has not been discovered [6,23]. Proteomic and transcriptome investigations, however, show that *T. gondii* and *N. caninum* differ greatly from one another in some characteristics. Although both species are tissue-dwelling parasites with many of the same traits as *T. gondii*, *N. caninum* does not infect humans and does not have the same variety of hosts. Instead, it has an astonishing affinity for highly effective vertical transmission in cattle. The variations in virulence, biology, and transmission routes between infection with *T. gondii* and infection with *N. caninum* may be more correlated with the high regulation of host protein phosphorylation and change of signaling pathways during *T. gondii* infection [21,24].

When *N. caninum* was analyzed immunologically, the results of the proteomic analysis show that at least 42 distinct protein spots of the organism responded to the anti-*N. caninum* serum, and at least 18 of those spots also responded to the anti-*T. gondii* serum. Additionally, the anti-*T. gondii* serum reacted with at least 31 protein locations of *T. gondii*, of which at least 19 protein sites did so with the anti-*N. caninum* serum [25]. The divergence of secreted virulence factors, notably rhoptry kinases, and an unanticipated expansion of surface antigen gene families were both displayed by *N. caninum*. In *N. caninum*, the rhoptry kinase ROP18 is pseudogenized, which may explain why *Neospora* cannot phosphorylate host immunity-related GTPases like *Toxoplasma* does. *Toxoplasma*’s pathogenicity is assumed to be largely dependent on this defense mechanism [21].

## 3. Protective Immunity against Toxoplasmosis and Neosporosis

Tachyzoites may be the first parasite stage to successfully interact with host immune effectors shortly after infection to establish the infection. *N. caninum* or *T. gondii* infection, however, can have a wide range of effects on the host depending on the host type, route of infection, physiological parameters (age, sex, pregnancy), and the parasite. Even in the same host with similar physiological conditions, infection-related symptoms might vary, providing more proof of the immune system’s critical function. Antigen-presenting cells (APCs), particularly macrophages and dendritic cells (DCs), as well as interferon-gamma (IFN-γ), which is integrated in the creation of high levels of pro-inflammatory mediators, are typically activated as part of the early immune response against such parasites. Tachyzoites swiftly develop into bradyzoites (dormant stage) in response to this inflammatory environment, allowing them to conceal themselves from host defenses by posing as immune effectors. The two arms of the immune response are the T helper 1 (Th1) and T helper 2 (Th2) subpopulations. Th1 cells emit IFN-γ, interleukin-2 (IL-2), IL-12, and tumor necrosis factor-alpha (TNF-α), while Th2 cells secrete IL-4, IL-5, IL-10, and IL-13. Th1 response and IFN-γ production are largely responsible for the protective immune response against toxoplasmosis and neosporosis [6,26,27,28].

A variety of immune cells induce IFN-γ production in response to macrophage IL-12 secretion. IFN-γ is a critical component of host resistance to *T. gondii* and *N. caninum* [29,30]. IFN-γ can activate macrophages and boost cytotoxic T cells, allowing them to eliminate intracellular parasites and infected cells. In more detail, IFN-γ exerts its anti-parasitic activity by increasing immunity-related GTPases (IRGs) and p65 guanylate-binding proteins (GBPs) to disrupt the parasitophorous vacuole [31], up-regulating indoleamine 2,3-dioxygenase for tryptophan depletion [32], producing nitric oxide (NO) by inducible NO synthase [33], activating P2X7 receptors [34], and increasing the release of oxygen radicals [35]. Furthermore, IFN-γ plays an important role in enhancing further IFN-γ production and additional cellular immune responses that favor pathogen killing and host protection via feedback mechanisms. This is achieved via the stimulation of T-helper and cytotoxic cells and the inhibition of T-regulatory cells (Figure 2).

In mice, the IRGs are particularly crucial for the manipulation of *T. gondii* and other intracellular infections [36,37]. IRGs are drawn to the vacuolar membrane in response to the detection of a PV, altering its structure and harming the released tachyzoites [38]. Another category of immune effectors, known as GBPs, is also elevated in response to IFN-γ. Instead of being drawn directly to the PV, GBPs gather close to the membrane, where they occupy groups of membrane vesicles [39]. Similarly, human cells use STAT1 and IFN-γ signaling to limit parasite replication in culture [40]. Although many of the interferon stimulated genes (ISGs) are similar in mouse and human, human cells control intracellular *T. gondii* by very different mechanisms than those described for mouse cells [41]. First, only two IRGs are expressed in humans; one is constitutively expressed in the testis (IRGC), and the other (IRGM) is truncated and probably does not function similarly to that of mice [42]. Second, even though human cells express a diverse range of GBPs [43], it has been questioned whether these proteins are involved in host defense. This is because a clustered regularly interspaced short palindromic repeats-associated protein 9 (CRISPR/CAS9) deletion of the locus containing the GBPs failed to demonstrate that these proteins are involved in the regulation of the intracellular replication of parasites in IFN-γ treated cells [44]. However, according to some investigations, some GBPs may help some human cells fight infection [45]. Additionally, these findings imply the existence of additional crucial pathways in human cells (Figure 2). In this regard, although there are differences between human and rodent toxoplasmosis, mouse models have already been identified and reviewed for researching several aspects of toxoplasmosis [41]. Rodents are natural hosts for the transmission of T. gondii and hence laboratory mice provide a reasonable model to study the immunological events involved in the control of this infection. Even for the clinical picture, various mouse models were used efficiently for studying different clinical forms of toxoplasmosis such as encephalitis, ocular form, and congenital infection [41].

The hyper-activation of the Th1 responses can also be harmful to the host milieu, where elevated levels of proinflammatory cytokines result in pathology. Lacking IL-10, the Th1 response modulator will aggravate the infection status and lead to greater immunopathology [46]. A previous report found that the IFN-γ-deficient mice that were susceptible to the *N. caninum* infection produced large quantities of IL-10, but resistant wild-type mice produced considerable levels of IFN-γ and IL-4 during the acute infection. An IFN-γ/IL-4 balance may be essential for regulating cellular responses to parasite invasion in light of the balance of Th1/Th2-type cytokine production [47]. In order to reduce inflammation, IL-27 also encourages regulatory T (Treg) cells, which restrict Th1 cell-mediated immunity [48]. Conversely, antibodies help to control infection in *T. gondii* and *N. caninum* infections by neutralising secreted antigens or preventing the spread of the parasite [27,29]. Purified peroxiredoxin 1 (IgG) antibodies inoculated intraperitoneally to *T. gondii*-infected SCID mice lessened the severity of the infection relative to unvaccinated control mice [49] (Figure 2).

The molecular interactions occurring at the interface between macrophages or dendritic cells (DCs) as antigen-presenting cells (APCs) and T cells, or what is known as the immunological synapse, play an important role in the immune response against infectious agents. Therefore, regulation of this interaction is considered a vital tool to improve the protection against pathogens without causing detrimental inflammation. Some of the molecular interactions affecting the fate of the APC-T cell interplay. T-cell receptor (TCR) binding to the major histocompatibility complex molecules (MHC) class I or II on the APC surface, which is responsible for the antigenic specificity on the surface of APC and T cells, and modulates APCs immunogenicity and T-cell function, respectively. In addition, clusters of differentiation CD-28, CD-80, CD-86, and CD-8 are critical for this specific interaction. Changes in the normal function of these molecules could lead to unbalanced APC-T-cell synapses that either cause a failure to control infections or cause immune overstimulation [27,29,50] (Figure 3).

Many vaccine trials against *T. gondii* and *N. caninum* infections have been conducted using various vaccine antigens, strategies, and animal models. However, no potent and safe vaccines have been commercialized on a large scale, and numerous obstacles are still hindering the development of vaccines that can completely prevent tissue cyst formation and/or fully block vertical transmission [28,29,30]. Regarding *Toxoplasma*, only Ovilis Toxovax, as a live attenuated S48 strain vaccine, has been commercially approved for use in sheep. This vaccine is available for *T. gondii* to prevent abortion in sheep and neonatal mortality in lambs. However, Toxovax has a relatively short shelf life and cannot fully eliminate the parasite. Furthermore, because the molecular basis of this live-attenuated tachyzoite S48 strain is not fully understood, spontaneous mutations may raise questions owing to the likelihood of reversion to a pathogenic phenotype, which can be extremely dangerous for those who are already at risk. Additionally, this vaccine (Toxovax) is pricey, has serious side effects, and is only accessible in Europe and New Zealand [30,51,52]. Despite the progress in finding an effective vaccine against *T. gondii*, no licensed vaccine is available for human application [53]. In a similar vein, the Bovilis Neoguard vaccine is the only anti-bovine neosporosis vaccine that has been produced and is approved for commercial use in cattle. It is made from the tachyzoite lysate antigen and has been utilized for many years in several nations. However, due to its ineffectiveness and potential negative effects on pregnant heifers that received the vaccine, this vaccine’s usage is now restricted [54].

Vaccine research against *T. gondii* or *N. caninum* initially focused on the use of live, attenuated, and killed tachyzoites. Due to recent significant advancements in the field of genetic engineering, several methods have been employed to generate attenuated vaccines. Numerous parasite strains that have undergone genetic modification through the deletion of specific genes have demonstrated a strong resistance to toxoplasmosis or neosporosis. The challenge dose, challenge period, parasite background, and host sensitivity all had a role in the protective effectiveness of altered parasites. But not all genetically altered parasites can lead to a robust immunity to a wild-type assault [6,30]. Moreover, their use is limited due to the risk of resumption of pathogenicity. Therefore, recent trends in vaccine development are shifting to subunit or vector-based vaccines. Recombinant proteins type of subunit vaccine that have shown effectiveness as vaccine antigens, either in a single formula or in combination with an adjuvant [28,29]. The recombinant protein is dependent on the utilization of a produced parasite antigen either in a prokaryotic or eukaryotic vector in each host cell in a former stage. As of late, a huge advance in the manufacturing of recombinant protein vaccines has been accomplished by utilizing adjuvant materials for designated vaccine antigens. The high safety of recombinant vaccine antigens is a fundamental advantage over any other type of vaccine, especially if they provide long-term and strong immunoprotective efficacy [6,28,55,56,57]. A brief description of recombinant protein preparation and evaluation as a vaccine antigen is illustrated in Figure 4. However, more detail on the used methods and materials can be found in our recently published chapter [58].

Additional steps have been conducted for entrapping the recombinant protein in liposomes coated with neoglycolipids containing oligo-mannose residues (OMLs). OML-enclosed target antigens are selectively taken up with the aid of using peripheral APCs, and then migrate to lymphoid tissues using a mannose receptor-dependent pathway. This adjuvant has proven its efficiency in improving the immunogenicity and protective efficacy of many recombinant antigens derived from *T. gondii* and *N. caninum* [29,59] (Figure 5).

Animals infected experimentally or naturally with *T. gondii* acquire an efficient immune protection for the next infection, which is not observed in the case of *N. caninum* infection, even though the level of protective antibodies in infected animals is high. Regarding *T. gondii*, there is a consensus for the induction of long-lasting protective immunity in infected hosts. This might be related to the induction of a vigorous innate and adaptive immune response manifested by a strong T cell immunity [60]. In the case of *N. caninum*, the protective immunity appears to be more effective in cows that are infected again from an exogenous source (oocysts) than in cows that relapse from an endogenous infection. Therefore, inducing protective immunity in cows that are already infected naturally is a problem [5]. This might be related to the differences in susceptible hosts, and to the lack of understanding of the host immune response against *N. caninum* because of the remarkably shorter time spent in research than that for *T. gondii*.

## 4. Role of Macrophages in Host-Parasite Interaction during Toxoplasmosis and Neosporosis

During early infection, macrophages constitute the first line of defense against almost all pathogens. Not only do macrophages have innate immunity, but they can also mediate the acquired or adaptive immunity either via a direct pathway through antigen processing, preparation, and presentation or an indirect pathway by secreting many effector molecules, including cytokines, chemokines, and other antimicrobial peptides. In addition, macrophages can perform their antiparasitic effect during the parasite killing through the phagocytosis process [61,62]. The early stage of macrophage activation in response to foreign antigens appears in its irregular shape and the large number of extensions or projections of the cell membrane, allowing the rapid capture of extracellular material. Added to structural changes, numerous functional changes also occur, including alterations in metabolism, reactive oxygen species levels, and the production of immune-related molecules [63,64].

Macrophages are skilled APCs that are vital for triggering immune reactions against the vaccine antigens or those liberated by the parasite upon natural infection. In such instances, various and numerous immune effector molecules are produced. In this review, we navigate an appropriate immunoscreening approach for the identification of potential vaccine candidates against *T. gondii* and *N. caninum*. Previous knowledge and our recent approach for vaccine development revealed a strong correlation between the host resistance against parasites or the efficacy of vaccine antigens and the macrophage secretion of IL-12 in murine models [6,28,29,55,56,57].

## 5. IL-12 Production from Macrophages during *T. gondii* or *N. caninum* Infection

Numerous studies revealed the potential of live parasites, whole antigens or a certain antigen of *T. gondii* or *N. caninum* in modulating the macrophage response toward the production of IL-12. Although these studies have not investigated the aspects of vaccine development, they corroborated the role of macrophage IL-12 in host resistance against either *T. gondii* or *N. caninum* infection [6,62]. In the same context, live tachyzoites of type I and type II strains of *T. gondii* could trigger the high production of IL-12 from cultured murine macrophage. The IL-12 release was significantly higher in type II than type I explaining the high resistance of the host against type II [65,66]. Additionally, using another experimental approach, the infection of mice per os via *T. gondii* cysts induced a high recruitment of numerous immune cells, including macrophages, in the intestinal cells of infected mice. High levels of IL-12p40 production and expression were also observed, which were primarily derived from macrophages rather than other immune cells. This IL-12p40 was essential for the induction of MyD88-independent type 1 immunity [67]. Not only the live parasite stages but also the treatment with tachyzoite lysate antigen (TLA) of *T. gondii* induced a significant increase in IL12p40 production in a dose-dependent manner in stimulated mouse macrophages in vitro [68].

Additionally, certain *T. gondii*-derived molecules demonstrated a similar effect of high productivity of macrophage IL-12. Recombinant protein TgCyp18 showed the potential to promote the macrophage production of IL-12p40 via a cysteine-cysteine chemokine receptor 5- (CCR5) dependent pathway [69]. Similarly, the treatment of bone marrow-derived macrophages (BMMs) with recombinant profilin-like protein in *T. gondii* (TgPLP) robustly elevated the production of IL-12 through a MyD88 signaling pathway [70]. The treatment of a naturally isolated murine macrophage or RAW 264.7 cell line with chemically synthesized or extracted *T. gondii* glycosylphosphatidylinositols (GPIs) resulted in higher IL-12 production than in non-treated cells [71,72]. In a more recent research trend, gene deletion, disruption, or mutation technology has also participated in the discovery and confirmation of the role of certain parasite molecules in host-parasite interaction. Most of these studies revealed the suppressive effect of abrogation of key molecules of *T. gondii,* such as GRA15, GRA24, and profilin-induced macrophage production of IL-12 against wild-type parasites [73,74,75,76]. Nevertheless, the disruption of other groups of *T. gondii* molecules revealed an opposite effect on IL-12 production from macrophages. Consistently, the IL-12 level culminated in the supernatant of the macrophage infected with *T. gondii* lacking either Rhoptry protein 16 (ROP16) [77] or ROP18 [78] against those of the wild type parasites. However, these results are consistent with other previous reports that highlighted the main function of ROP antigens as virulence factors that the parasites use to halt the host immunity, reviewed by Bradley and Sibley, (2007) [79]. A summary of previous studies assessed the effect of *T. gondii* on macrophage production of IL-12 is shown in Table 1.

In the case of *N. caninum*, the arsenals and strategies of battle in host-parasite interaction are very similar to those that occurred with *T. gondii*. Many studies have confirmed the efficacy of macrophage modulation via *N. caninum*. To figure out such effects, live *N. caninum* tachyzoites (Nc-1) and macrophages of mouse origin were mostly used. In this context, Abe et al. (2014) [80] found that infection of mice with Nc-1 tachyzoites induced obvious recruitment of macrophages to the site of infection and was associated with increased IL-12p40 in cultured macrophages ex vivo. Similarly, stimulation of murine macrophages in vitro with Nc-1 tachyzoites was associated with IL-12 release. This effect was aggravated after inhibition of certain pathways such as p38 MAPK [81], Dectin [82], or TLR11/MEK/ERK [83]. In another study, the immunoprotective role of IL-12 was emphasized in vivo, as mice treated with rIL-12 were less susceptible to encephalitis and showed lower brain parasite load at 3 wpi with *N. caninum* than the non-treated mice [84]. Even in cattle, the highly susceptible animal to *N. caninum,* cultured macrophages infected with Nc-Spain1H exhibited excessive ROS release and IL12p40 expression in comparison to cells infected with Nc-Spain7, and both strains were higher than non-infected cells [23]. Accordingly, macrophages of different phenotypes performed similar immune responses against *N. caninum* infection, either of murine or bovine cell origin.

Additionally, exploiting the research progress in *T. gondii*, recent experimental approaches have identified certain *N. caninum*-derived effector molecules against macrophage promotion and IL-12 production. Similar to preparation techniques for TgGPIs, chemically synthesized NcGPI induced IL-12 production by a cultured mouse macrophage cell line and from natural DCs of mouse origin [85]. With a different experimental approach to antigen preparation, rNc14-3-3 activates the MAPK and AKT signaling pathways, accompanied with excessive production of IL-12p40 from cultured peritoneal macrophages in vitro collected from mice [86]. Nishikawa et al. (2018) [87] investigated the role of various *N. caninum* effector molecules using CRISPR/CAS9. This study focused on assessing the role of several dense granule proteins, including NcGRA6, NcGRA7, NcGRA14, and cyclophilin. Mice infected with NcGRA7KO parasites exhibited higher survival rates in mice of different genetic backgrounds (BALB/c, C57BL/6, and TLR2KO) than other KO or wild-type parasites. Interestingly, IL-12p40 production in cultured macrophages infected with wild-type or NcGRA7-complemented parasites were higher than that in infected with NcGRA7KO parasites. However, many other inflammatory mediators were also induced in wild-type parasites such as IL-6, TNF-α, IFN-γ, CCL2, and CXCL10, suggesting the cause of the severity of infection with Nc-1 compared to NcGRA7KO parasites [87]. Thus, we can deduce the intricate milieu of the immune system and host-parasite interaction. A summary of previous studies assessed the effect of *N. caninum* on macrophage production of IL-12 is shown in Table 2.

Based on our previous studies, IL-12 production from macrophages is a promising approach for the selection of potential vaccine candidates against *T. gondii* and *N. caninum* infections. These studies demonstrated recombinant proteins of *T. gondii* peroxiredoxin 1 (TgPrx1) and TgPrx3, and *N. caninum* dense granule protein 6 (NcGRA6) and cyclophilin-entrapped in oligo-mannose coated-liposomes (NcCyp-OML) as potential vaccine candidates [49,57,88,89]. This effect starts with antigen uptake through appropriate receptors and proceeds by activation of related signaling pathways depending on the activated gateway. The TLR/MyD88-NF-κB axis normally produces IL-12, which is responsible for IFN-γ production and Th1-mediated immunity [6]. Furthermore, previous research established the importance of macrophages as a first line of defense and IL-12 as a regulatory cytokine for IFN-γ production. This potential triad of macrophage/IL-12/IFN-γ is critical for combating many invading infectious agents, such as *T. gondii* and *N. caninum*, and thus for vaccine development [6,62]. As a preliminary approach, vaccine antigens should possess the competency to promote macrophage production of IL-12. However, this competence will also require the successful interaction with other host molecules and the induction of a proper immune response, which is prioritized over a strong immune response. In our recent report, we provided a detailed description for the preparation and stimulation of macrophages from mouse peritoneum [58]. This approach indicates the ease, practicability, and cost-effectiveness of vaccine antigen testing. Therefore, macrophage stimulation evidenced in IL-12 production can be used as a reliable method for assessment of vaccine efficacy before further evaluation using in vivo experiments, which will reduce the time, effort, and expenses required for such research trials. However, some vaccine antigens can stimulate a Th2 immune response, resulting in anti-inflammatory IL-10 production and antibody-mediated protective immunity. This effect may be regulated through IL-4/IL-13 receptors and STAT6 pathways [90] (Figure 6).

## 6. Macrophage Role in Vaccine Development against *T. gondii* and *N. caninum*

Concerning *T. gondii*, we found that the stimulation of mouse peritoneal macrophages with TgPrx1 [49] and TgPrx3 [57], revealed the robust production of IL-12. Both recombinant proteins of TgPrx1 and TgPrx3 conferred strong and moderate protection in mice, respectively. Similarly, NcGRA6 [88] and NcCyp-OML [89] promoted the mouse macrophage and triggered IL-12 production. Regarding protective potentials, both antigens substantially protected the immunized mice. Noteworthy, the four above-mentioned vaccine antigens showed potential to stimulate humoral and cellular immunities in comparison to control groups (Table 3).

Ex vivo addition of relevant antigens to vaccinated groups’ spleen cells increased cell proliferation and IFN-γ production, indicating that such antigens can elicit cellular immunity. In addition, low levels of IL-4 were detected in TgPrx1 and TgPrx3, but not in NcGRA6 or NcCyp-OML-immunized mice. The induced higher levels of IFN-γ compared to IL-4 in all the above-mentioned antigens indicate that the triggered immunity was biased toward Th1 immunity via the IL-12/IFN-γ axis (Table 3). Therefore, we can deduce that macrophage promotion proven in IL-12 production may be used as a helpful approach for forecasting vaccine effectualness before more analysis mistreatment in vivo experiments [49,57,88,89].

Although numerous studies have been conducted to investigate the interaction of the parasite or its derived molecules with macrophages, the relationship between macrophages and vaccine development has not been thoroughly investigated in previous reports on our tested parasites. Another study found a strong association between *N. caninum* antigens as vaccine candidates and dendritic cells (DCs) stimulation. This study revealed the stimulation of DCs by ROP2 and ROP40 and protein disulfide isomerase (PDI) in mixing with TLR2 ligand. However, the formulation of TLR2 ligand with ovalbumin as a non-specific antigen did not confer protection against *N. caninum* infection in mice, emphasizing the contribution of *N. caninum* antigens in the protective mechanism [91].

In the same manner, a previous study showed that TLR2 is critical for the protective immunity of the NcCyp-OML vaccine against *N. caninum* infection in mice [89]. This protection was abrogated when using NcCyp or OML adjuvant with *N. caninum* non-specific antigen (Glutathione-*S*- transferase) alone. These results suggest the importance of TLR2 as a pattern recognition receptor (PRR) in vaccine-induced protection. Other studies revealed that APCs, which include macrophages and DCs, additionally make contributions in inducing immunity against infectious agents via means of imparting antigens to MHC classes I and II, resulting in lymphocyte activation and subsequently antigen-specific acquired immunity [92]. This effect is mostly initiated and regulated by PPRs via TLRs, which are highly expressed by APCs. Previous research found that mice lacking TLR2 were more susceptible to *T. gondii* infection than wild-type mice [93] and *N. caninum* [94]. Furthermore, TLR2 was involved in both innate and adaptive immunity against *N. caninum* in murine cells [95]. The activation of the TLR2/NF-κB axis pathway is strongly correlated with protection against microbial infections [96,97]. In addition, previous studies revealed the efficient immunomodulatory effect of recombinant protein antigens derived from *T. gondii* and *N. caninum* in the early infection of mice or murine-derived cells. In mice, rTgCyp18 ligation with CCR5 causes macrophages to release nitric oxide (NO), IL-12, and TNF-γ [69]. Additionally, rTgPrx treatment diminished caspase-1 function and IL-1β productivity, with a concurrent elevation of IL-10 release in the case of treatment of bone-marrow-derived macrophages from mice [98]. In the case of *N. caninum,* rNcCyp caused murine and bovine cells to migrate in a CCR5-dependent manner [99]. These data suggest the efficiency of recombinant protein in triggering an immune response from effector immune cells.

Testing the cellular immune response rather than the humoral response for vaccine candidates seems to be very crucial for intracellular protozoan parasites including *T. gondii* and *N. caninum*. However, vaccination of animals/humans associated with the production of protective antibodies against the pathogen is crucial for long-lasting immunity. Consequently, in our previous studies summarized in Table 1, humoral immune response was also investigated. As noticed for either *T. gondii* [49,57] or *N. caninum* [88,89], all the tested vaccine antigens could generate IgG1 production after immunization and before the challenge infection, suggesting the development of humoral or antibody-based immunity and its significance in induction of protective immunity. One approach to ascertain the contribution of humoral immunity is to carry out passive protection experiments with immune sera or purified specific antibodies. Indeed, the passive immunization and treatment of mice with purified polyclonal antibodies of TgPrx1 improved the immune response of treated mice against *T. gondii* infection compared to the control group that received PBS, although it was not significant [49].

Opsonization, complement activation, neutralisation of released antigens, and antibody-dependent cellular cytotoxicity are the traditional methods of antibody-mediated defense. However, the predicted outcome for providing protective immunity against studied vaccine antigens, particularly for intracellular parasites, is also the synergistic effect of humoral and cellular immunity. Other antibody-mediated defense systems recently revealed may play a key role in limiting intracellular infections. The intracellular pathogens’ surface-bound antibodies may cause transcriptional reactions that could disrupt the physiology of the microbes. Additionally, the presence of a particular antibody can enhance protection by moderating the inflammatory response by triggering FcR, activating complement, and neutralizing immunomodulatory microbial products. These changes in the inflammatory response change the release of inflammatory mediators [100].

## 7. Concluding Remarks

Vaccination with recombinant protein as a vaccine antigen is remarkably increasing because of safety and ease of preparation. However, the immune response against most kinds of vaccine antigens is either short-term or inadequate. Thus, formulation with appropriate adjuvant or preparation as cocktail antigens are recently sought to overcome the demerits of immunological inadequacy. Another alternative option is the discovery of vaccine antigens that possess adjuvant activity evidenced in APCs’ stimulation and exertion of specific, potent, and long-lasting immunity. Under the current situation of no potent vaccines or efficient treatment, we provided an important insight for in vitro screening of potential vaccine candidates against both *T. gondii* and *N. caninum* infection. The data presented in this review summarize promising results regarding the usefulness of macrophage stimulation based on IL-12 production in the assessment of potent vaccine candidates against *T. gondii* and *N. caninum* infection. Vaccine antigens of TgPrx1, TgPrx3, NcGRA6, and NcCyp are revealed as potent vaccine candidates after preliminary tests using the macrophage stimulation approach. Additionally, many previous reports also demonstrated the importance of macrophage IL-12 production in resistance against *T. gondii* and *N. caninum* infection. This data can be exploited for the improvement of vaccine development by reducing the required time and costs of the vaccine evaluation process for such parasites and other infectious agents. This study is also consistent with the recent strict ethical considerations of minimizing the use of experimental animals via avoiding their useless testing. However, further studies are required to test a macrophage-based system on different kinds of vaccine antigens and various parasites. Simultaneously, the assessment of different immune effector cells as DCs or molecules as NF-κB might be useful.

## Figures and Tables

**Figure 1 vaccines-10-02082-f001:**
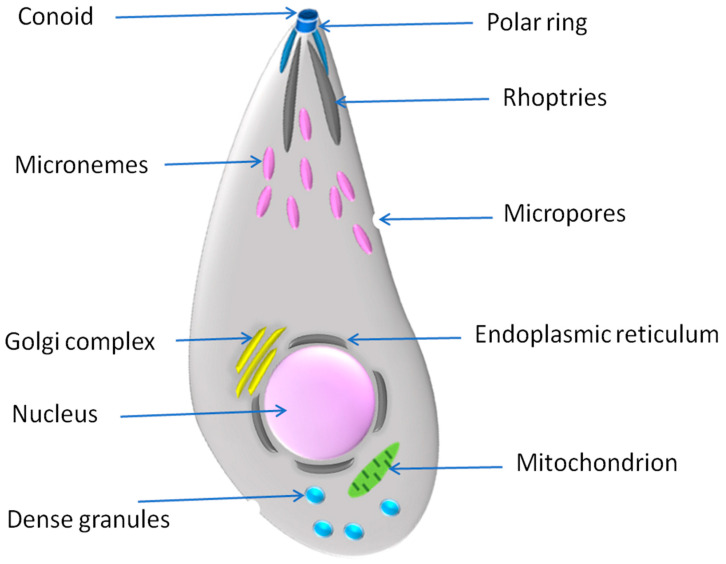
Ultrastructure showing an illustration for the common vital cell organelles in the tachyzoite of *T. gondii* and *N. caninum*.

**Figure 2 vaccines-10-02082-f002:**
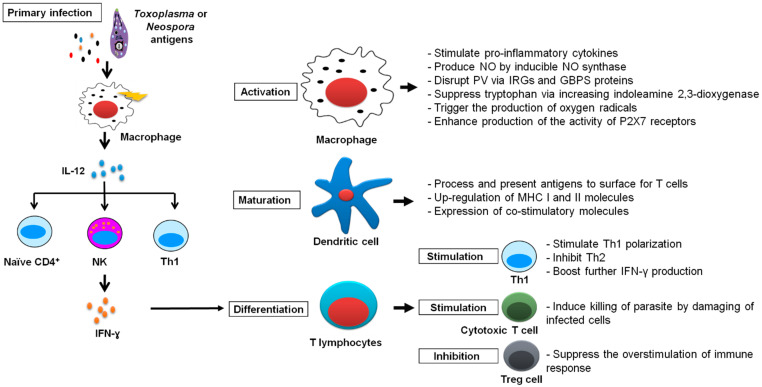
The role of IFN-γ in protection against *T. gondii* and *N. caninum*. Upon infection with *T. gondii* or *N. caninum,* macrophages as a first defense line secrete IL-12, which is a key factor to produce IFN-γ from T-helper and natural killer cells. IFN-γ interacts with IFN-γ-producing cells, such as macrophages, dendritic cells and T-cells. IFN-γ stimulates macrophages for combating intracellular pathogens through numerous pathways. Also, it enhances dendritic cells for MHC I and II up-regulation. For T-cells, IFN-γ interacts with T-cells to stimulate their differentiation toward the Th1 subset. Through positive feedback, IFN-γ enhances its own production in Th1 cells and inhibits Th2 and Th17 differentiation. IFN-γ is required for maturation of naïve T-cells to effector CD8+ T-cells. The IFN-γ is the main cytotoxic molecule released by these cells. Additionally, the immunosuppressive T regulatory cells are inhibited by IFN-γ.

**Figure 3 vaccines-10-02082-f003:**
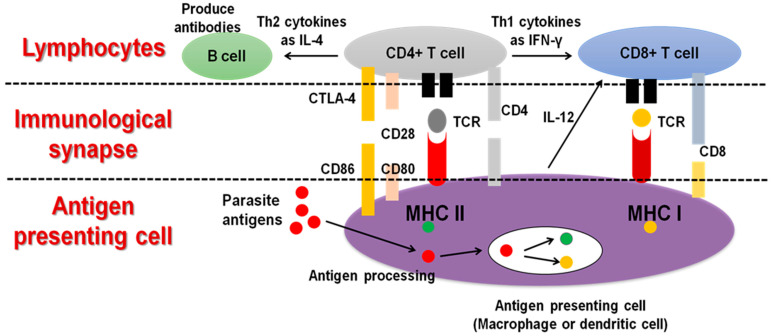
Immunological synapse between antigen presenting cells and lymphocytes. In case of infection and antigen release, antigen presenting cells (APCs), mostly DCs or macrophages, will act for antigen processing and presentation for triggering the subsequent proper immune response. The molecular interactions occurring at the interface between APC and T cells is determined by the type of pathogen and stage of infection and will determine the subsequent infection. The molecular interactions include T-cell receptor (TCR) and cluster of differentiation CD-28, CD-80, CD-86 and CD-8 for binding to the MHC class I or II on the APC surface and can determine the activation of T-helper (CD4+) and T-cytotoxic cells (CD8+), respectively.

**Figure 4 vaccines-10-02082-f004:**
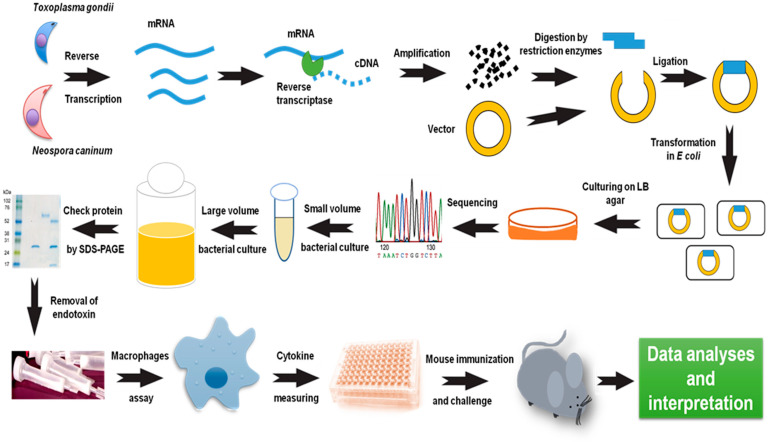
Main steps of vaccine antigen preparation and evaluation. Recombinant protein is prepared based on the cDNA library via synthesis of cDNA from mRNA using reverse transcriptase. Then, the target is amplified using specific primers and conventional polymerase chain reaction. Molecular cloning of the target gene is started by digestion using restriction enzymes, including the target vectors (e.g., pGEX-4T or pET). After ligation, the designed vector with the gene is inserted in cloning *Escherichia coli* (*E. coli*) as DH5α and plated on luria bertani agar. Positive colonies are confirmed by sequencing and then transformed into protein expression *E. coli* BL21. Positive colonies are tested for protein expression using a small volume culture. Protein expression is started by culturing on a large volume culture (0.5–1 L). Recombinant protein purity and quantity is tested via running on SDS-PAGE. Endotoxin is removed from the obtained protein before in vitro testing on macrophages and in vivo assessment in mice. Successful vaccine antigen can be predicted through macrophage production of IL-12 and confirmed via prolongation of mouse survival after the challenge with a parasite.

**Figure 5 vaccines-10-02082-f005:**
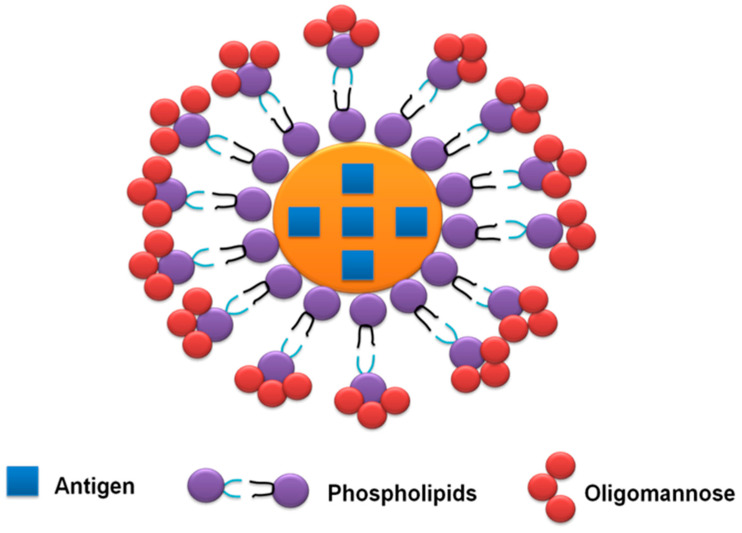
Representation of oligomannose-coated liposome adjuvant containing vaccine antigen.

**Figure 6 vaccines-10-02082-f006:**
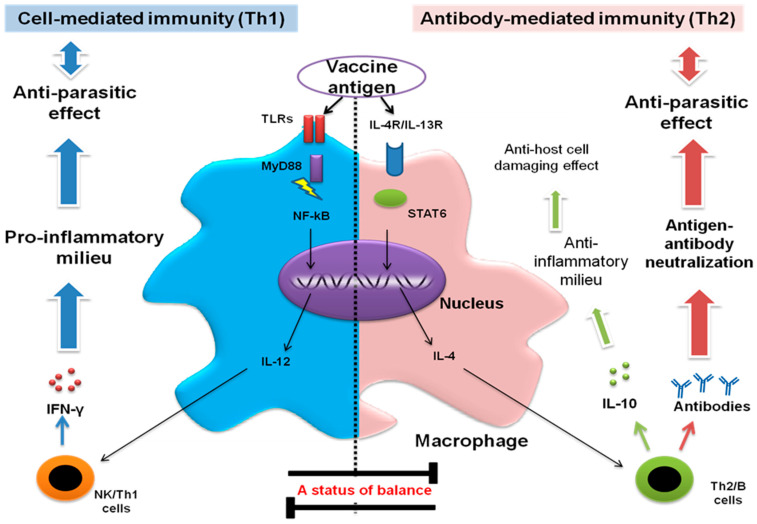
Contribution of macrophages in host protective immunity. Macrophages are the main source of IL-12 production. This cytokine is critical for IFN-γ production. TLRs, MyD88 and NF-kB are the efficient signaling pathway cascade toward the elimination of *T. gondii* or *N. caninum* via stimulating IFN-γ production from NK and Th1 cells. The positive feedback among IL-12 and IFN-γ result in the stimulation of Th1 cells and formation of IgG2 antibody, the main effectors for cell mediated immunity. In addition, Th2 mediated immunity participates in protective immunity. The secretion of IL-4 as feedback for IL-10 secretion also contributes to protective immunity either by secretion of regulatory (IL-10) or anti-inflammatory (IL-4) cytokines or through the specific antibody production. The balance between Th1 and Th2 immunity is important to prevent host cell damage from the cytokine storm effect or the excessive parasite proliferation.

**Table 1 vaccines-10-02082-t001:** A summary for previous studies assessed the effect of *T. gondii* or its derived molecules on macrophage production of IL-12.

*Toxoplasma* Antigen	Macrophage Stimulation	References
Tachyzoites of type I virulent and type II avirulent strains	Tachyzoites of type II strain induced high macrophage IL-12 release than type I.	[65,66]
Cyst of ME49 strain	Oral *T. gondii* infection induced high recruitment of macrophages and IL-12p40 production in the lamina propria of infected mice via MyD88-independent type 1 immunity.	[67]
Tachyzoite lysate antigen (TLA) from RH strain	The treatment with TLA increased IL12p40 in a dose-dependent manner.	[68]
Cyclophilin 18 (TgCyp18)	Recombinant TgCyp18 increased production of IL-12 via CCR5 dependent pathway.	[69]
Profilin-like protein in *T. gondii* (TgPLP)	TgPLP treatment induced the excessive release of IL-12 dependent on MyD88 signaling pathway in BMMs.	[70]
Glycosylphosphatidylinositols (TgGPIs)	High IL-12 production in RAW 264.7 or thioglycolate elicited mouse macrophage cells treated with chemically synthesized TgGPIs.	[71,72]
Dense granule 15 (TgGRA15)	TgGRA15 is responsible for the induction of IL-12 secretion by infected mouse macrophages evidenced using KO parasites.	[74]
TgGRA24	TgGRA24 is obviously assuming a significant part in inducing IL-12p40 synthesis, as its absence seriously compromises both IL-12 expression and liberation in mouse BMDM.	[76]
Profilin (TgPF)	TgPF is required for recognition by macrophage TLR11 and TLR12 and subsequent production of IL-12.	[73,75]
Rhoptry protein 16 (TgROP16)	TgROP16 in type I parasites impairs Stat3 activation and decreased IL-12 production as evidenced using wild type and KO parasites.	[77]
TgROP18	IL-12 production was higher in supernatant of RAW264.7 cells infected with TgROP18 mutant or KO parasites than those of wild type associated with inhibition of host NF-kB.	[78]

**Table 2 vaccines-10-02082-t002:** A summary for previous studies assessed the effect of *N. caninum* or its derived molecules on macrophage production of IL-12.

*Neospora* Antigen	Macrophage Stimulation	References
*N. caninum* tachyzoites (Nc-1)	*N. caninum* triggered the obvious recruitment of macrophages at the scene of infection with concurrent elevated IL-12p40 release in infected murine macrophages in vitro.	[80]
Live tachyzoites and soluble extract of *N. caninum* (Nc-1)	Cultured mouse macrophages showed an elevated IL-12p40 production when stimulated by live tachyzoites and antigen extracts, which was dependent on the p38 MAPK pathway.	[81]
*N. caninum* tachyzoites (Nc-1)	In vitro, IL-12p40 was elevated in *N. caninum* infected macrophages of Dectin-1^−/−^ mice than WT.	[82]
*N. caninum* tachyzoites (Nc-1)	*N. caninum* infection induced an obvious increase in IL-12p40 by mouse macrophages in vitro. This level was markedly decreased after abolishing the TLR11/MEK/ERK pathway.	[83]
*N. caninum* tachyzoites (Nc-1)	The treatment of *N. caninum* infected mice with rIL-12 greatly relieved encephalitis and reduced the parasite burden in the mouse brain.	[84]
*N. caninum* tachyzoites (Nc-Spain7 and Nc-Spain1H)	Cultured bovine macrophages showed higher ROS and IL-12p40 liberations when infected with Nc-Spain1H than those infected with Nc-Spain7.	[23]
Glycosylphosphatidylinositols (NcGPIs)	Chemically synthesized NcGPIs induced IL-12, TNF-α, and IL-1β secretion by a mouse cell line of macrophages and natural dendritic cells.	[85]
*N. caninum* 14-3-3 protein (Nc14-3-3)	Recombinant Nc14-3-3 activates the AKT and MAPK and signaling pathways, and elevated IL-12p40 production from mouse peritoneal macrophages.	[86]
NcGRA7	Infection of mouse macrophages with NcGRA7KO parasites induced lower secretion of IL-12p40 than those of parental or complemented parasites.	[87]

**Table 3 vaccines-10-02082-t003:** Summary of previous reports on *T. gondii* and *N. caninum* vaccine antigens and their effects on macrophages.

Vaccine Antigen	Innate Immunity	Humoral Immunity	Cellular Immunity	Protection	References
TgPrx1	Stimulated IL-12 production from naturally isolated mouse macrophages and IL-6 from RAW cell line	Generated high IgG1 and low IgG2 levels after the third immunization	Triggered IFN-γ production from splenocytes of immunized mice	Survival rate in immunized mice was higher (66.7%) than the control group that received PBS (27.8%)	[49]
TgPrx3	Stimulated IL-12 production from naturally isolated mouse peritoneal macrophages	Generated high IgG1 and IgG2 levels after the third immunization	Triggered IFN-γ production from splenocytes of immunized mice	Immunized mice showed higher survival (55.6%) than the control group that received PBS (27.8%)	[57]
NcGRA6	Induced excessive IL-12 liberation from naturally isolated mouse peritoneal macrophages	Generated only IgG1 after the third immunization	Triggered IFN-γ production from splenocytes of vaccinated mice	Higher survival rate was observed in vaccinated mice (91.7%) than the control group that received PBS (16.7%)	[88]
NcCyp-OML	Induced excessive IL-12 liberation from naturally isolated mouse peritoneal macrophages	Generated only IgG1 after the third immunization	Triggered IFN-γ production from splenocytes of vaccinated mice	Higher survival rate was noticed immunized mice (83.3%) than the control group that received PBS (16.7%)	[89]

## Data Availability

Not applicable.

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
