# Peer review of "Exploiting the Macrophage Production of IL-12 in Improvement of Vaccine Development against Toxoplasma gondii and Neospora caninum Infections"

_vaccines, 2022, doi:10.3390/vaccines10122082_

Round 1

Reviewer 1 Report

This seems to be an extension of your work published as reference 56 - so it is a question whether this adds significantly to our body of knowledge.

And, most critically, as you say in line 163/4 - "human cells control intracellular T. gondii by very different mechanisms than those described for mouse cells"

The bovine Neospora vaccines has been discontinued for more than ten years now.

The English needs a lot of attention, very colloquial in parts

Author Response

General author response

We are greatly appreciating the comments from editor and all reviewers. In this revision note, we have answered to comments through point by point revision. We have indicated author response in this revision note under title of Author’s response. Also, we indicated our changes and revisions in manuscript as blue-colored fonts. Numbers of citations and references have been changed because of references added based on reviewers’ comments. In addition, as a response to reviewer comment, we have created another file with newly added or deleted or modified information as tracking changes option. Hopefully, our revision and answers in below would be sufficient for publication of our study in your prestigious MDPI vaccines journal.

Point by point revision

Reviewer 1

Comments and Suggestions for Authors

- This seems to be an extension of your work published as reference 56 - so it is a question whether this adds significantly to our body of knowledge.

Author’s response

- We apologize for this confusion, although similar concept, not only the type of article is different but also the content of both studies (current and above-mentioned reference no. 58, in current new version) are highly different. Our previous study (Ref. no. 58) is a chapter that published in a book series and only focused in providing detailed illustrations for materials and methods that have been used. the current one provided plenty of evidences for usefulness of our approach oppositely to another one that focused on methodology.

- As a literature review, in addition to our study (ref. 56), we also collected high number of other relevant studies and reviews that supported the current hypothesis. These observations were described in details our current study in sections of discussion and conclusive points, citation by numerous literatures, and illustration in numerous tables and figures to increase visualization.

- Moreover, under current situation of no potent vaccines or efficient treatment (as already mentioned in our study), we provided an important insight for in vitro screening of potential vaccine candidates against both Toxoplasma gondii and Neospora caninum infection with providing numerous evidences and elucidating mechanisms. This approach will be highly beneficial in this aspect because it will reduce the cost, effort and time spent in further assessments of prepared vaccine antigens using experimental animals.

- Actually, not only our additional study, but also several studies are required to conceptualize this approach because of the expected high importance in improvement of vaccine development in such tested parasites and might be of wider spectrum vaccine research.

- This study also is consistent with the recent strict ethical considerations of minimizing the use of experimental animals via avoiding their useless testing.

- The concept of this study is deduced from our previous experience (about 10 years) in doing experiments and research in such research field, as noticed in the following relevant publications

Original papers

1-      Ragab M. Fereig, Hanan H. Abdelbaky, Yasuhiro Kuroda, Yoshifumi Nishikawa. Critical role of TLR2 in triggering protective immunity with cyclophilin entrapped in oligomannose-coated liposomes against Neospora caninum infection in mice. Vaccine. January 2019, 37, 937-944. (Original research article, 5-year impact factor: 3.309)

2-      Ragab M. Fereig, Naomi Shimoda, Hanan H. Abdelbaky, Yasuhiro Kuroda, Yoshifumi Nishikawa. Neospora GRA6 possesses immune-stimulating activity and confers efficient protection against Neospora caninum infection in mice. Veterinary Parasitology. February 2019, 267, 61-68. (Original research article, 5-year impact factor: 2.434)

3-      Ragab M. Fereig, Hanan H. Abdelbaky, Yoshifumi Nishikawa. Vaccination with Neospora GRA6 interrupts the vertical transmission and partially protects dams and offspring against Neospora caninum infection in mice. Vaccines. February 2021, 9: 155. 0.3390/vaccines9020155. (Original research article, 5-years impact factor: 5.513)

4-      Ragab M. Fereig, Yasuhiro Kuroda, Mohamad Alaa Terkawi, Motamed Elsayed Mahmoud, Yoshifumi Nishikawa. Immunization with Toxoplasma gondii peroxiredoxin 1 induces protective immunity against toxoplasmosis in mice. PLOS ONE. April 2017, 12(4):e0176324.) (Original research article, 5-year impact factor: 3.352)

5-      Ragab M. Fereig, Yoshifumi Nishikawa. Peroxiredoxin 3 promotes IL-12 production from macrophages and partially protects mice against infection with Toxoplasma gondii. Parasitology International, 65, 741-748. December 2016. (Original research article, IF: 1.929)

6-      Ragab M. Fereig, Yoshifumi Nishikawa. Genetic Disruption of Toxoplasma gondii peroxiredoxin (TgPrx) 1 and 3 Reveals the Essential Role of TgPrx3 in Protecting Mice from Fatal Consequences of Toxoplasmosis. International Journal of Molecular Sciences. March 13, 2022, 23, 3076. https:// doi.org/10.3390/ijms23063076 (Original paper; 5-years impact factor: 6.132)

7-      Fumiaki Ihara, Ragab M Fereig, Yuu Himori, Kyohko Kameyama, Kosuke Umeda, Sachi Tanaka, Rina Ikeda, Masahiro Yamamoto, Yoshifumi Nishikawa. Toxoplasma gondii dense granule proteins 7, 14, and 15 are involved in modification and control of the immune response mediated via NF-κB pathway. Frontiers in Immunology. July 2020, 11:1709 (Original research article, 5-year impact factor: 5.733)

8-      Yoshifumi Nishikawa, Naomi Shimoda, Ragab M. Fereig, Tomoya Moritaka, Kousuke Umeda, Maki Nishimura, Fumiaki Ihara, Kaoru Kobayashi, Yuu Himori, Yutaka Suzuki,  Hidefumi Furuoka. Neospora caninum dense granule protein 7 regulates pathogenesis of neosporosis by modulating host immune response. Applied and Environmental Microbiology, 84, pii: e01350-18. July 2018. (Original research article, 5-year impact factor: 4.272)

Review article

9-      Ragab M. Fereig, Yoshifumi Nishikawa. From signaling pathways to distinct immune responses: Key factors for establishing or combating Neospora caninum infection in different susceptible hosts. Pathogens. May 2020 16;9(5):384. (Review article, 5-years impact factor: 4.066)

10-  Ragab M. Fereig, Hanan H. Abdelbaky, Adel Elsayed Ahmed Mohamed, Yoshifumi Nishikawa. Recombinant subunit vaccines against Toxoplasma gondii: Successful experimental trials using recombinant DNA and proteins in mice in a period from 2006 to 2018. Journal of Veterinary Medicine and Animal Sciences. 2018; 1: 1005. (Review article).

Chapters in book

11-  Ragab M. Fereig, Yoshifumi Nishikawa. Towards a preventive strategy for toxoplasmosis: current trends, challenges, and future perspectives for vaccine development. Methods in Molecular Biology, 1404. 153-164. April 2016.

12-  Ragab M. Fereig, Yoshifumi Nishikawa. Macrophage stimulation as a useful approach for immunoscreening of potential vaccine candidates against Toxoplasma gondii and Neospora caninum infections. In: Thomas S. (eds) Vaccine Design. Methods in Molecular Biology, vol 2411. Humana, New York, NY. https://doi.org/10.1007/978-1-0716-1888-2_8. November 2021.

Patents under final judgments

13-  Name: Vaccine Adjuvant

Application number: 2017-252032

Title: Vaccination with dense granule 6 protects the mice against the experimental lethal and transplacental infection with Neospora caninum.

Inventors: Yoshifumi Nishikawa, Ragab M. Fereig

14-  Name: Vaccine development against Neospora caninum infection

Application number: 2018-71224

Title: Vaccination with oligomannose-coated liposome-entrapped Neospora caninum cyclophilin protects the mice against the experimental lethal infection with Neospora caninum

Inventors: Yoshifumi Nishikawa, Ragab M. Fereig

Relevant information was already added in the current version of manuscript for highlighting the importance of this study as follows

Newly added information:

“Under the current situation of no potent vaccines or efficient treatment, we provided an important insight for in vitro screening of potential vaccine candidates against both T. gondii and N. caninum infection.” (page 16, lines 579-581).

“This study also is consistent with the recent strict ethical considerations of minimizing the use of experimental animals via avoiding their useless testing.” (page 16, lines590-592)

Previously added information

“This valuable information might help vaccinologists in developing potent vaccines against different pathogens in a considerably shorter time and with less effort than traditional procedures. In addition, this information may encourage other researchers to develop different immunoscreening methods based on other immune effector cells or molecules.” (page 2, lines 61-65)

“This approach indicates the ease, practicability, and cost-effectiveness of vaccine antigen testing. Therefore, macrophage stimulation evidenced in IL-12 production can be used as a reliable method for assessment of vaccine efficacy before further evaluation using in vivo experiments, which will reduce the time, effort, and expenses required for such research trials.” (page 13, lines 468-472)

- And, most critically, as you say in line 163/4 - "human cells control intracellular T. gondii by very different mechanisms than those described for mouse cells"

Author’s response

Although we totally agree with the reviewer 1 comment, the immunity against T. gondii is still ambiguous and many secrets are still uncovered yet. For examples, immune response against T. gondii not only differed between human and rodents, but also within different human individual itself. Also, since the clinical presentation of infection with T. gondii is diverse, there is not a single animal model that addresses all clinical presentations observed in humans. However, in this review, we mostly focused on role of macrophage production of IL-12 in vaccine development not only in T. gondii but also in N. caninum research fields. In which, mouse are the most commonly used animal model for such purposes and the target of our study is the establishment of potential immunoscreening approach (ex vivo method) for reduction time, cost, and efforts in assessment of vaccine candidates at the research level prior to application in susceptible animals including human. In addition, in our current studies, we also demonstrated the efficacy of our tested vaccine antigens (TgPrx1 and TgPrx3) in stimulating human effector immune molecules (unpublished).

Relevant information has been added as follows;

“In this regard, although there are differences between human and rodent toxoplasmosis, mouse models have already been identified and reviewed for researching several aspects of toxoplasmosis [41]. Rodents are natural hosts for transmission of T. gondii and hence laboratory mice provide a reasonable model to study the immunological events involved in the control of this infection. Even for clinical picture, various mouse models were used efficiently for studying different clinical forms of toxoplasmosis such as encephalitis, ocular form, and congenital infection [41]”. (page 5, lines 209-216)

- The bovine Neospora vaccines has been discontinued for more than ten years now.

Author’s response

We agree with reviewer 1 comment. However, as this review focused on vaccine development against T. gondii and N. caninum, we added some information on the previous vaccine trials for farm animals, although of limited success. In addition, we also referred on the termination of their use currently with reporting the causes as follow:

“Despite the progress in finding an effective vaccine against T. gondii, no licensed vaccine is available for human application [53]. In a similar vein, Bovilis Neoguard vaccine is the only anti-bovine neosporosis vaccine that has been produced and is approved for commercial use in cattle. It is made from tachyzoite lysate antigen and has been utilized for many years in several nations. However, due to its ineffectiveness and potential negative effects on pregnant heifers that received the vaccine, this vaccine's usage is now restricted [54].” (page 7, lines 278-284).

- The English needs a lot of attention, very colloquial in parts

Author’s response

English editing and grammar check are revised by many professional online programs, and additionally some language revisions were applied again to avoid repeating (marked by blue colored fonts).

Reviewer 2 Report

Thank you for the opportunity to review this manuscript. In this manuscript entitled “Exploiting the macrophage production of IL-12 in improvement of vaccine development against Toxoplasma gondii and Neospora caninum infections”, the authors discuss about the immunological screening method for antigen screening to predict protozon vaccine outcome special attention of IL-12 production in macrophages. This review article could be helpful for the vaccine researchers. This manuscript is well written and scientifically sounds correct. Additionally, this manuscript contains valuable information for readers. However, this manuscript needs to be clarified before publication in Viruses listed below.

Major comments:

1.      In the section 6, I was not sure about the correlation between the level of IL-12 production from macrophages with the addition of antigen and the outcome of vaccine efficacy, therefore I think additional sentences would be needed.

Minor comments:

2.      Line 45: Toxoplasma gondii and neospora caninum should be spelled out here.

3.      Line 66: Please correct “Toxoplasma gondii” to “T. gondii”.

4.      Line 97: N. caninum should be regular font.

5.      Line 141: IFN and IL should be spelled out here.

6.      Line 266: Please correce “cNDA” to “cDNA”.

7.      Line 269: E. coli should be spelled out here.

8.      Line 271: E. coli should be italic.

9.      Line 271: Please correct to E. coli BL21.

Author Response

Manuscript ID: vaccines-1967348

Title

Exploiting the macrophage production of IL-12 in improvement of vaccine development against Toxoplasma gondii and Neospora caninum infections.

General author response

We are greatly appreciating the comments from editor and all reviewers. In this revision note, we have answered to comments through point by point revision. We have indicated author response in this revision note under title of Author’s response. Also, we indicated our changes and revisions in manuscript as blue-colored fonts. Numbers of citations and references have been changed because of references added based on reviewers’ comments. In addition, as a response to reviewer comment, we have created another file with newly added or deleted or modified information as tracking changes option. Hopefully, our revision and answers in below would be sufficient for publication of our study in your prestigious MDPI vaccines journal.

Point by point revision

Reviewer 2

Comments and Suggestions for Authors

- Thank you for the opportunity to review this manuscript. In this manuscript entitled “Exploiting the macrophage production of IL-12 in improvement of vaccine development against Toxoplasma gondii and Neospora caninum infections”, the authors discuss about the immunological screening method for antigen screening to predict protozoan vaccine outcome special attention of IL-12 production in macrophages. This review article could be helpful for the vaccine researchers. This manuscript is well written and scientifically sounds correct. Additionally, this manuscript contains valuable information for readers. However, this manuscript needs to be clarified before publication in Vaccines listed below.

Author’s response

We greatly appreciate the reviewer 2 positive comments and hopefully our answers and responses in the current version of manuscript will be sufficient for acceptance of publication.

- Major comments:

- In the section 6, I was not sure about the correlation between the level of IL-12 production from macrophages with the addition of antigen and the outcome of vaccine efficacy, therefore I think additional sentences would be needed.

Author’s response

In this section (Macrophage role in vaccine development against T. gondii and N. caninum), we aimed to present our experience in vaccine research against such parasites with special reference to the macrophage role. Noticeably, a limited number of relevant researches have been focused on such aspect that was exclusively to our lab (Table 3), despite numerous research studies on vaccine development for these parasites. In this regard and to expand this point, other non-vaccine studies that provided indirect evidence for vaccine efficacy-IL-12 production from macrophages were also added and discussed in this study as illustrated in table (1, 2).

Additional information have been added to this section

“Testing cellular immune response rather than humoral one for vaccine candidates seems to be very crucial for intracellular protozoan parasites including T. gondii and N. caninum. However, vaccination of animals / humans associated with the production of protective antibodies against the pathogen is crucial for long-lasting immunity. Consequently, in our previous studies summarized in table 1, humoral immune response was also investigated. As noticed, either for T. gondii [49, 57], or for N. caninum [88, 89], all the tested vaccine antigens could generate IgG1 production after immunization and before challenge infection suggesting the development of humoral or antibody-based immunity and its significance in induction of protective immunity. One approach to ascertain the contribution of humoral immunity is to carry out passive protection experiments with immune sera or purified specific antibodies. Indeed, the passive immunization and treatment of mice with purified polyclonal antibodies of TgPrx1 improved the immune response of treated mice against T. gondii infection compared to the control group received PBS, although it was not significant [49].

Opsonization, complement activation, neutralisation of released antigens, and antibody-dependent cellular cytotoxicity are the traditional methods of antibody-mediated defence. However, the predicted outcome for providing protective immunity against studied vaccine antigens, particularly for intracellular parasites, is also the synergistic effect of humoral and cellular immunity. Other antibody-mediated defense systems recently revealed may play a key role in limiting intracellular infections. The intracellular pathogens' surface-bound antibodies may cause transcriptional reactions that could disrupt the physiology of the microbes. Additionally, the presence of a particular antibody can enhance protection by moderating the inflammatory response by triggering FcR, activating complement, and neutralizing immunomodulatory microbial products. These changes in the inflammatory response change the release of inflammatory mediators [100]. “ (page 15, lines 545-570). 

- Minor comments:

  1. Line 45: Toxoplasma gondii and neospora caninum should be spelled out here.

Author’s response

It is modified to Toxoplasma gondii (T. gondii) and Neospora caninum (N. caninum). (page 2, lines 45-46 )

  1. Line 66: Please correct “Toxoplasma gondii” to “T. gondii”.

Author’s response

We have used in this form because it is in beginning of sentence and paragraph. (page 2, lines 68 )

  1. Line 97: N. caninum should be regular font.

Author’s response

Corrected (page 3, line 101 )

  1. Line 141: IFN and IL should be spelled out here.

Author’s response

Done at first appearance in the text

interferon-gamma (IFN-γ) (page 4, line 169)

interleukin-2 (IL-2) (page 4, lines 174-175)

  1. Line 266: Please correce “cNDA” to “cDNA”.

Author’s response

Corrected (page 8, line 308)

  1. Line 269: E. coli should be spelled out here.

Author’s response

Spelled out Escherichia coli (E. coli) (page 8, lines 311-312)

  1. Line 271: E. coli should be italic.

Author’s response

Corrected (page 8, line 313)

  1. Line 271: Please correct to E. coli BL21.

Author’s response

Corrected (page 8, line 313)

Reviewer 3 Report

My main concern is that Authors published already very similar article entitled “Macrophage Stimulation as a Useful Approach for Immunoscreening of Potential Vaccine Candidates Against Toxoplasma gondii and Neospora caninum Infections” as a chapter in Methods in Molecular Biology, which is cited by the Authors.

It would be good to mention that dogs and cats are definitive hosts for those parasites. The immune response depends on the stage of the parasite as well as the location in host organism.

It is true that both T. gondii and N. caninum are very similar due to the species are closely related, however they differ in antigenic compounds and immune response reviled in infected animals. Animals infected experimentally or naturally with Toxoplasma gondii acquire an immune protection for the next infection, which is not observed in case of Neospora  caninum infection even though the level of protective antibodies in infected animals is high.

The idea of testing cellular response instead of humoral for vaccine candidates seems to be very interesting, however usually in the idea of vaccination of animals / humans production of protective antibodies against the pathogen is crucial for ling lasting immunity.

Minor errors:

in the Keywords: Toxoplasma is missing

in the Introduction line 66 Toxoplasma gondii and Neospora caninum - full names of the parasites should be used in the beginning

Author Response

Manuscript ID: vaccines-1967348

Title

Exploiting the macrophage production of IL-12 in improvement of vaccine development against Toxoplasma gondii and Neospora caninum infections.

General author response

We are greatly appreciating the comments from editor and all reviewers. In this revision note, we have answered to comments through point by point revision. We have indicated author response in this revision note under title of Author’s response. Also, we indicated our changes and revisions in manuscript as blue-colored fonts. Numbers of citations and references have been changed because of references added based on reviewers’ comments. In addition, as a response to reviewer comment, we have created another file with newly added or deleted or modified information as tracking changes option. Hopefully, our revision and answers in below would be sufficient for publication of our study in your prestigious MDPI vaccines journal.

Point by point revision

Reviewer 3

Comments and Suggestions for Authors

- My main concern is that Authors published already very similar article entitled “Macrophage Stimulation as a Useful Approach for Immunoscreening of Potential Vaccine Candidates Against Toxoplasma gondii and Neospora caninum Infections” as a chapter in Methods in Molecular Biology, which is cited by the Authors.

Author’s response

- We apologize for this confusion, although similar concept, not only the type of article is different but also the content of both studies (current and above-mentioned reference no. 56) are highly different. Our previous study (Ref. no. 58 in current new version) is a chapter that published in a book series and only focused in providing detailed illustrations for materials and methods that have been used. the current one provided plenty of evidences for usefulness of our approach oppositely to another one that focused on methodology.

- As a literature review, in addition to our study (ref. 58), we also collected high number of other relevant studies and reviews that supported the current hypothesis. These observations were described in details our current study in sections of discussion and conclusive points, citation by numerous literatures, and illustration in numerous tables and figures to increase visualization.

- Moreover, under current situation of no potent vaccines or efficient treatment (as already mentioned in our study), we provided an important insight for in vitro screening of potential vaccine candidates against both Toxoplasma gondii and Neospora caninum infection with providing numerous evidences and elucidating mechanisms. This approach will be highly beneficial in this aspect because it will reduce the cost, effort and time spent in further assessments of prepared vaccine antigens using experimental animals.

- Actually, not only our additional study, but also several studies are required to conceptualize this approach because of the expected high importance in improvement of vaccine development in such tested parasites and might be of wider spectrum vaccine research.

- This study also is consistent with the recent strict ethical considerations of minimizing the use of experimental animals via avoiding their useless testing.

- The concept of this study is deduced from our previous experience (about 10 years) in doing experiments and research in such research field, as noticed in the following relevant publications

Original papers

1-      Ragab M. Fereig, Hanan H. Abdelbaky, Yasuhiro Kuroda, Yoshifumi Nishikawa. Critical role of TLR2 in triggering protective immunity with cyclophilin entrapped in oligomannose-coated liposomes against Neospora caninum infection in mice. Vaccine. January 2019, 37, 937-944. (Original research article, 5-year impact factor: 3.309)

2-      Ragab M. Fereig, Naomi Shimoda, Hanan H. Abdelbaky, Yasuhiro Kuroda, Yoshifumi Nishikawa. Neospora GRA6 possesses immune-stimulating activity and confers efficient protection against Neospora caninum infection in mice. Veterinary Parasitology. February 2019, 267, 61-68. (Original research article, 5-year impact factor: 2.434)

3-      Ragab M. Fereig, Hanan H. Abdelbaky, Yoshifumi Nishikawa. Vaccination with Neospora GRA6 interrupts the vertical transmission and partially protects dams and offspring against Neospora caninum infection in mice. Vaccines. February 2021, 9: 155. 0.3390/vaccines9020155. (Original research article, 5-years impact factor: 5.513)

4-      Ragab M. Fereig, Yasuhiro Kuroda, Mohamad Alaa Terkawi, Motamed Elsayed Mahmoud, Yoshifumi Nishikawa. Immunization with Toxoplasma gondii peroxiredoxin 1 induces protective immunity against toxoplasmosis in mice. PLOS ONE. April 2017, 12(4):e0176324.) (Original research article, 5-year impact factor: 3.352)

5-      Ragab M. Fereig, Yoshifumi Nishikawa. Peroxiredoxin 3 promotes IL-12 production from macrophages and partially protects mice against infection with Toxoplasma gondii. Parasitology International, 65, 741-748. December 2016. (Original research article, IF: 1.929)

6-      Ragab M. Fereig, Yoshifumi Nishikawa. Genetic Disruption of Toxoplasma gondii peroxiredoxin (TgPrx) 1 and 3 Reveals the Essential Role of TgPrx3 in Protecting Mice from Fatal Consequences of Toxoplasmosis. International Journal of Molecular Sciences. March 13, 2022, 23, 3076. https:// doi.org/10.3390/ijms23063076 (Original paper; 5-years impact factor: 6.132)

7-      Fumiaki Ihara, Ragab M Fereig, Yuu Himori, Kyohko Kameyama, Kosuke Umeda, Sachi Tanaka, Rina Ikeda, Masahiro Yamamoto, Yoshifumi Nishikawa. Toxoplasma gondii dense granule proteins 7, 14, and 15 are involved in modification and control of the immune response mediated via NF-κB pathway. Frontiers in Immunology. July 2020, 11:1709 (Original research article, 5-year impact factor: 5.733)

8-      Yoshifumi Nishikawa, Naomi Shimoda, Ragab M. Fereig, Tomoya Moritaka, Kousuke Umeda, Maki Nishimura, Fumiaki Ihara, Kaoru Kobayashi, Yuu Himori, Yutaka Suzuki,  Hidefumi Furuoka. Neospora caninum dense granule protein 7 regulates pathogenesis of neosporosis by modulating host immune response. Applied and Environmental Microbiology, 84, pii: e01350-18. July 2018. (Original research article, 5-year impact factor: 4.272)

Review article

9-      Ragab M. Fereig, Yoshifumi Nishikawa. From signaling pathways to distinct immune responses: Key factors for establishing or combating Neospora caninum infection in different susceptible hosts. Pathogens. May 2020 16;9(5):384. (Review article, 5-years impact factor: 4.066)

10-  Ragab M. Fereig, Hanan H. Abdelbaky, Adel Elsayed Ahmed Mohamed, Yoshifumi Nishikawa. Recombinant subunit vaccines against Toxoplasma gondii: Successful experimental trials using recombinant DNA and proteins in mice in a period from 2006 to 2018. Journal of Veterinary Medicine and Animal Sciences. 2018; 1: 1005. (Review article).

Chapters in book

11-  Ragab M. Fereig, Yoshifumi Nishikawa. Towards a preventive strategy for toxoplasmosis: current trends, challenges, and future perspectives for vaccine development. Methods in Molecular Biology, 1404. 153-164. April 2016.

12-  Ragab M. Fereig, Yoshifumi Nishikawa. Macrophage stimulation as a useful approach for immunoscreening of potential vaccine candidates against Toxoplasma gondii and Neospora caninum infections. In: Thomas S. (eds) Vaccine Design. Methods in Molecular Biology, vol 2411. Humana, New York, NY. https://doi.org/10.1007/978-1-0716-1888-2_8. November 2021.

Patents under final judgments

13-  Name: Vaccine Adjuvant

Application number: 2017-252032

Title: Vaccination with dense granule 6 protects the mice against the experimental lethal and transplacental infection with Neospora caninum.

Inventors: Yoshifumi Nishikawa, Ragab M. Fereig

14- Name: Vaccine development against Neospora caninum infection

Application number: 2018-71224

Title: Vaccination with oligomannose-coated liposome-entrapped Neospora caninum cyclophilin protects the mice against the experimental lethal infection with Neospora caninum

Inventors: Yoshifumi Nishikawa, Ragab M. Fereig

- It would be good to mention that dogs and cats are definitive hosts for those parasites.

Author’s response

This information is already modified and reported as follow:

“Dogs as definitive host for N. caninum, and cats; the definitive host for T. gondii are critical for the transmission and epidemiology of neosporosis and toxoplasmosis, respectively, among other susceptible animals…...” (page 2, lines 83-86)

- The immune response depends on the stage of the parasite as well as the location in host organism.

 Author’s response

“Tachyzoites may be the first parasite stage to successfully interact with host immune effectors shortly after infection in an effort to establish the infection. N. caninum or T. gondii infection, however, can have a wide range of effects on the host depending on the host type, route of infection, physiological parameters (age, sex, pregnancy), and the parasite. Even in the same host with similar physiological conditions, infection-related symptoms might vary, providing more proof of the immune system's critical function. Antigen-presenting cells (APCs), particularly macrophages and dendritic cells (DCs), as well as interferon-gamma (IFN-γ), which is integrated in the creation of high levels of pro-inflammatory mediators, are typically activated as part of the early immune response against such parasites. Tachyzoites swiftly develop into bradyzoites (dormant stage) in response to this inflammatory environment, allowing them to conceal themselves from host defenses by posing as immune effectors” (page 4, lines 162-173)

“Proteomic and transcriptome investigations, however, have shown that T. gondii and N. caninum differ greatly from one another in a number of characteristics. Although both species are tissue-dwelling parasites with many of the same traits as T. gondii, N. caninum does not infect humans and does not have the same variety of hosts. Instead, it has an astonishing affinity for highly effective vertical transmission in cattle. The variations in virulence, biology, and transmission routes between infection with T. gondii and infection with N. caninum may be more correlated with the high regulation of host protein phosphorylation and change of signaling pathways during T. gondii infection [21, 24].” (page 3, 139-147).

- It is true that both T. gondii and N. caninum are very similar due to the species are closely related, however they differ in antigenic compounds and immune response reviled in infected animals.

Author’s response

“When N. caninum was analyzed immunologically, the results of the proteomic analysis showed that at least 42 distinct protein spots of the organism responded to the anti-N. caninum serum, and at least 18 of those spots also responded to the anti-T. gondii serum. Additionally, the anti-T. gondii serum reacted with at least 31 protein locations of T. gondii, of which at least 19 protein sites did so with the anti-N. caninum serum [25]. The divergence of secreted virulence factors, notably rhoptry kinases, and an unanticipated expansion of surface antigen gene families were both displayed by N. caninum. In N. caninum, the rhoptry kinase ROP18 is pseudogenized, which may explain why Neospora cannot phosphorylate host immunity-related GTPases like Toxoplasma does. Toxoplasma's pathogenicity is assumed to be largely dependent on this defense mechanism [21].” (page 3,4 , lines 148-157)

- Animals infected experimentally or naturally with Toxoplasma gondii acquire an immune protection for the next infection, which is not observed in case of Neospora caninum infection even though the level of protective antibodies in infected animals is high.

Author’s response

“Animals infected experimentally or naturally with T. gondii acquire an efficient immune protection for the next infection, which is not observed in case of N. caninum infection even though the level of protective antibodies in infected animals is high. Regarding T. gondii, there is a consensus for the induction of long-lasting protective immunity in infected hosts. This might be related to the induction of a vigorous innate and adaptive immune response manifested by a strong T cell immunity [60]. In case of N. caninum, the protective immunity appears to be more effective in cows that are infected again from an exogenous source (oocysts) than in cows that relapse from an endogenous infection. Therefore, inducing protective immunity in cows that are already infected naturally is a problem [5]. This might be related to the differences in susceptible hosts, and to the lack of understanding of the host immune response against N. caninum because of remarkable shorter time spent in research than that for T. gondii.” (page 9, lines 329-340)

- The idea of testing cellular response instead of humoral for vaccine candidates seems to be very interesting, however usually in the idea of vaccination of animals / humans production of protective antibodies against the pathogen is crucial for ling lasting immunity.

Author’s response

We totally agree to this comment and we already discussed this aspect in different occasions in the current version of manuscript.

“Conversely, antibodies help to control infection in T. gondii and N. caninum infections by neutralising secreted antigens or preventing the spread of the parasite [27, 29]. Purified peroxiredoxin 1 (IgG) antibodies inoculated intraperitoneally to T. gondii-infected SCID mice lessened the severity of the infection relative to unvaccinated control mice [49] (Figure 2)” (page 5, lines 226-230)

“However, some vaccine antigens can stimulate Th2 immune response, resulting in anti-inflammatory IL-10 production and antibody-mediated protective immunity. This effect may be regulated through IL-4/IL-13 receptors and STAT6 pathways [90] (Figure 6).” (page 13, lines 472-475)

In addition, we added this part to ascertain the importance of antibody-based immunity.

“Testing cellular immune response rather than humoral one for vaccine candidates seems to be very crucial for intracellular protozoan parasites including T. gondii and N. caninum. However, vaccination of animals / humans associated with the production of protective antibodies against the pathogen is crucial for long-lasting immunity. Consequently, in our previous studies summarized in table 1, humoral immune response was also investigated. As noticed, either for T. gondii [49, 57], or for N. caninum [88, 89], all the tested vaccine antigens could generate IgG1 production after immunization and before challenge infection suggesting the development of humoral or antibody-based immunity and its significance in induction of protective immunity. One approach to ascertain the contribution of humoral immunity is to carry out passive protection experiments with immune sera or purified specific antibodies. Indeed, the passive immunization and treatment of mice with purified polyclonal antibodies of TgPrx1 improved the immune response of treated mice against T. gondii infection compared to the control group received PBS, although it was not significant [49].

Opsonization, complement activation, neutralisation of released antigens, and antibody-dependent cellular cytotoxicity are the traditional methods of antibody-mediated defence. However, the predicted outcome for providing protective immunity against studied vaccine antigens, particularly for intracellular parasites, is also the synergistic effect of humoral and cellular immunity. Other antibody-mediated defense systems recently revealed may play a key role in limiting intracellular infections. The intracellular pathogens' surface-bound antibodies may cause transcriptional reactions that could disrupt the physiology of the microbes. Additionally, the presence of a particular antibody can enhance protection by moderating the inflammatory response by triggering FcR, activating complement, and neutralizing immunomodulatory microbial products. These changes in the inflammatory response change the release of inflammatory mediators [100].” (page 15, lines 570 )

Minor errors:

- In the Keywords: Toxoplasma is missing

Author’s response

 Added as “T. gondii” (page 15, line 570)

- In the Introduction line 66 Toxoplasma gondii and Neospora caninum - full names of the parasites should be used in the beginning

Author’s response

Full names were used (page 2, line 68)

Round 2

Reviewer 3 Report

Dear Authors, Thank you for your answers.

Minor editing/ spelling corrections are required.